# Clinical outcomes of circumcisions and prevalence of complications of male circumcisions: A five-year retrospective analysis at a teaching hospital in Ghana

**Sylvester Appiah Boakye**[1], **Frank Obeng**[1,2]*

**1** School of Medicine, University of Health and Allied Sciences, Ho, Ghana, **2** Department of Surgery, Urology Unit, University of Health and Allied Sciences, Ho, Ghana

* fobeng@uhas.edu.gh

## Abstract

Male circumcision is increasing in popularity due to its medical benefits, including reducing HIV prevalence. It is commonly performed by both health and non-health professionals, with most circumcisions occurring during the neonatal period. Studies suggest the benefits outweigh the risks, though complications can occur. This study aimed to determine the clinical outcomes of circumcisions and the prevalence of adverse events of circumcision in the Volta region of Ghana. A five-year retrospective descriptive and analytic study was conducted at Ho Teaching Hospital, using a structured data extraction sheet to collect demographic, clinical, and circumcision-related data from 186 cases. Among 186 circumcision cases, 23 (12.37%) experienced complications, with the most common being partial circumcision (43.48%), post-circumcision bleeding (21.74%), and urethrocutaneous fistula and/or wrongfully circumcised congenital hypospadias (13.04%). Low heamoglobin levels and infections were also noted. A significant relationship was found between the circumcision provider and complication rates (Chi-square = 16.975, p = 0.00). Doctors conducting circumcision had the lowest complication rates (4.3%), while nurses and traditional circumcisers had higher complication rates (39.1% and 34.8%, respectively). Circumcision-Revision surgery was the most common salvage surgery for circumcision mishaps (31.82%), with urethroplasty and hypospadias repair (for wrongful circumcised neonates born with hypospadias) accounting for 15.91%. Meatoplasties, glansplasties, fistulectomy plus primary repair and chordae-release surgeries were also performed. The success rate for salvage surgeries (first attempt) was 70%. Prompt initial management strategies were significantly associated with good outcomes. Under less-trained hands, circumcision could be catastrophic. Salvage surgeries for circumcision mishaps are associated with less favourable outcomes in about one-third of the cases, suggesting that circumcision mishaps are better prevented than salvaged. Training, guidance, and policy interventions are needed to reduce the incidence of circumcision-related mishaps. Public health campaigns to dissuade non-surgeon circumcisers to refrain from circumcising children with hypospadias but refer them, are urgently needed.

**Data availability statement:** Dataset has been uploaded in submission uploads as 'dataset' under supporting information.

**Funding:** The authors received no specific funding for this work.

**Competing interests:** The authors have declared that no competing interests exist.

## Author summary

This study aimed to assess the clinical outcomes of circumcisions and the prevalence of adverse events in the Volta region of Ghana. A retrospective analysis of 186 circumcision cases conducted at Ho Teaching Hospital over five years revealed that 12.37% of cases experienced complications. The most common complications included partial circumcision, post-circumcision bleeding, and urethrocutaneous fistula or mis-circumcised congenital hypospadias. The study found a significant relationship between the circumcision provider and complication rates, with doctors having the lowest complication rates (4.3%), compared to nurses (39.1%) and traditional circumcisers (34.8%). Salvage surgeries were required in 12.37% of cases, with circumcision-revision surgery being the most common. Other corrective procedures included urethroplasty and hypospadias repair. The success rate for these salvage surgeries was 70%. The study concluded that circumcision mishaps are better prevented than treated, with less trained practitioners contributing to higher complication rates. It emphasized the need for better training, policy interventions, and public health campaigns to discourage non-surgeons from performing circumcisions, particularly in cases involving hypospadias.

## Introduction

Male circumcision is a widely practiced surgical procedure, particularly in regions with religious, cultural, and public health imperatives [1]. In Ghana, the practice is almost universal, with a significant portion of the male population undergoing the procedure [2]. Despite its recognized benefits, including a reduction in the transmission of HIV and other sexually transmitted infections [3–5], circumcision is not without risks. Complications, though generally infrequent, can range from mild to severe and may include infections, excessive bleeding, and, in rare cases, permanent damage to penile structures [6]. Understanding the clinical outcomes and the frequency and nature of these complications is crucial for improving patient safety and refining surgical techniques.

Circumcision, a widely practiced procedure, has been extensively studied with regard to its techniques, benefits, and associated complications [7]. We give a brief background on the prevalence of circumcision complications, common types of complications, the relative rates of circumcision mishaps amongst the various providers of the service (circumcisers), initial management of complications, and the success rate these management strategies.

### Prevalence of circumcision complications

Circumcision prevalence varies globally, with Israel reporting 91.7% and Honduras less than 1%. Africa sees a 62% prevalence, with Ghana at 91.6% [4]. The prevalence of complications is influenced by factors like anatomical abnormalities, age, surgical technique, and medical comorbidities [8] In the U.S., complication rates are less than 0.5% [9], while in rural Ghana, the complication rate among infants is 8.1% [10]. These figures highlight the variability in circumcision complication rates worldwide and suggest a link between a country's development status and complication prevalence.

### Common circumcision complications

Complications from circumcision are well-documented, with haemorrhage/bleeding being the most common, occurring in 11.9% of cases in the United States of America. [3,11]. Other

complications include meatal stenosis, infection, oedema, penile hematoma, and urethrocutaneous fistula, among others. These complications are classified as early or late [6] and can vary by region. For example, in Ghana, urethrocutaneous fistula is the most common complication, followed by glans amputations and iatrogenic hypospadias [12]. Circumcision-related complications are rife in Ghana usually because most procedures are performed by less skilled providers [12,13].

This study aims to provide a comprehensive retrospective analysis of male circumcision outcomes over the past five years at Ho Teaching Hospital in the Volta Region. By identifying patterns and prevalence of complications, this research will contribute to better clinical practices and inform public health strategies in the region.

**The objectives of this study were**

- To evaluate the clinical outcomes of male circumcisions performed at Ho Teaching Hospital over the past five years.

- To identify and categorize the complications associated with male circumcision in this population.

- To assess the factors contributing to successful outcomes and complications.

This study's conceptual framework (see Fig 1) revolves around the relationship between circumcision techniques, healthcare provider experience, and patient outcomes. It posits that the clinical outcomes of circumcision are influenced by multiple factors, including the method used, the provider's skill level, and the age of the patient at the time of the procedure [14]. The framework will analyse these factors to determine their impact on the incidence of complications and overall surgical success.

## Materials and methods

This study was a five-year retrospective descriptive analysis of circumcisions at Ho Teaching Hospital (HOTH), focusing on clinical outcomes, prevalence of complications, and management of circumcision disasters. Data was collected from the children's ward, male ward, the child welfare clinics, as well as theatre where circumcisions and salvage surgeries for circumcision adverse events, were performed. Data on acute presentations of circumcision complications was also assessed from emergency room records (see S1 Data).

The study population included circumcision cases managed at HOTH and complication referrals from other health facilities within the Volta region. All circumcised children attending the child welfare clinic and cases with circumcision complications referred to HOTH between January 1, 2019, and December 31, 2023, were included. Cases with incomplete or duplicated data were excluded (S1 Data).

The study used a census approach [15], relying on secondary data from HOTH's electronic records and archives. A structured MS Excel data extraction tool was employed, covering patient demographics, clinical presentation, laboratory parameters, and outcomes. Data was collected through personal visits to hospital wards, ensuring completeness and confidentiality.

## Statistical analysis

Data analysis involved quantitative methods, including descriptive statistics, chi-square tests for categorical data and regression analysis for parametric data, using SPSS-version 25. All analyses were conducted at a 95% confidence level with a 5% significance threshold.

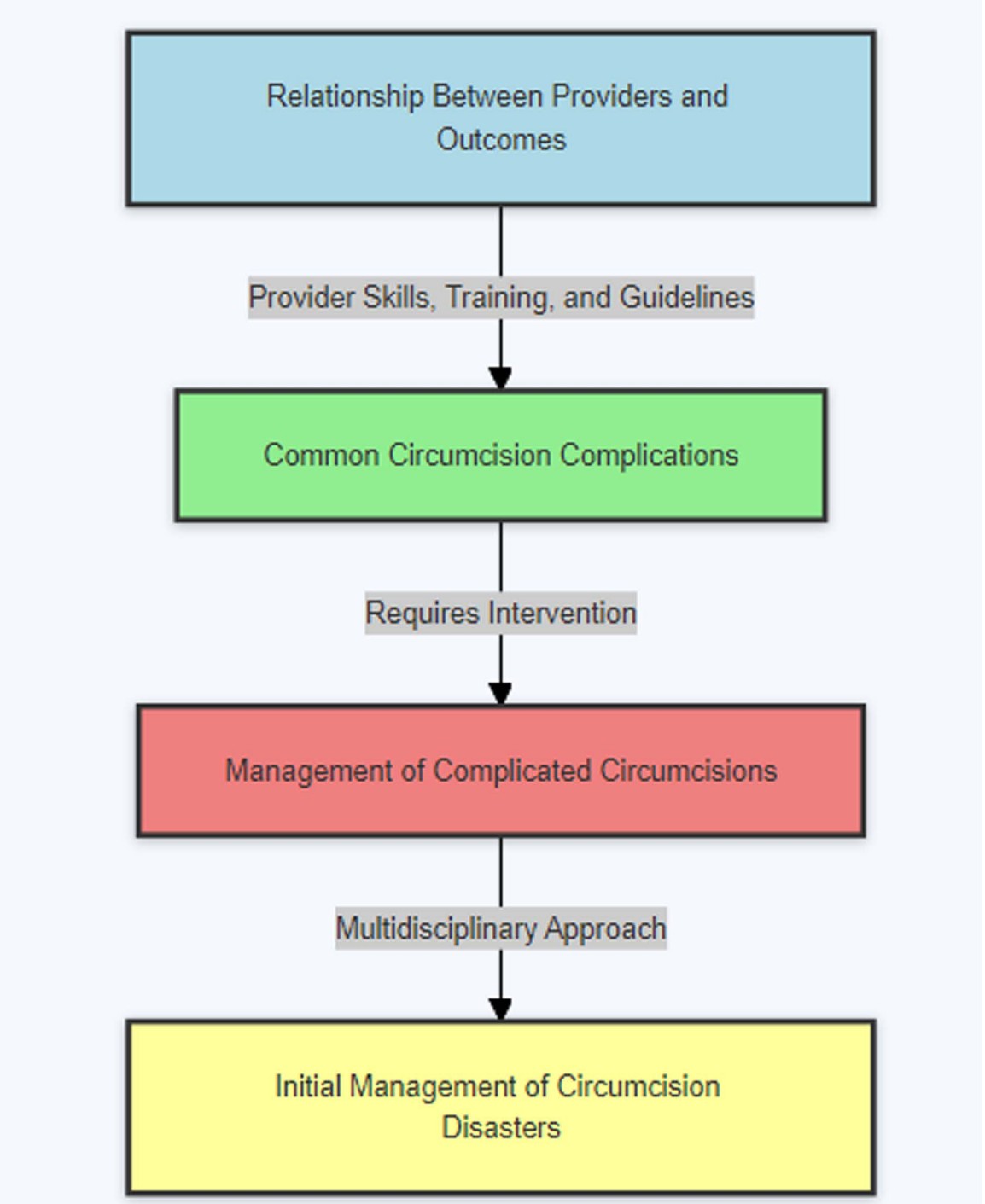

**Fig 1. Conceptual framework diagram.** Factors influencing circumcision outcomes based on a literature review.Source: Author's Construct based on Literature Review, 2024.

## Results

### Trends of circumcisions performed over the past 5 years

Fig 2 shows a trend of circumcision cases recorded annually from 2019 to 2023 in the study area. The graph indicates a steady increase in the number of circumcision cases over this period. Starting in 2019, the cases were below 20, and a gradual rise was observed in 2020, where cases slightly increased to around 20. From 2021 to 2022, the number of cases increased moderately, surpassing 40 in 2022. A sharp rise was noted between 2022 and 2023, where the cases jumped significantly, reaching a peak of about 80 by 2023. The green dotted line shows an upward trajectory of significant and consistent growth in the number of circumcision cases over the five years.

### Trends of salvage surgeries performed for referred circumcision mishaps over the past 5 years within the teaching hospital

Fig 3 revealed that over the past five years, there has been a steady increase in the number of cases documented. In 2019, there were 8 cases, accounting for 13.33% of the total. The number of cases slightly decreased in 2020, with 7 cases (11.67%), but increased again in 2021 to 9 cases (15.00%). The trend continued upward in 2022, with 13 cases (21.67%). The highest number of cases was recorded in 2023, with 22 cases, representing 36.67% of the total. Given that we know the total number of circumcision complications over the period to be 23, the

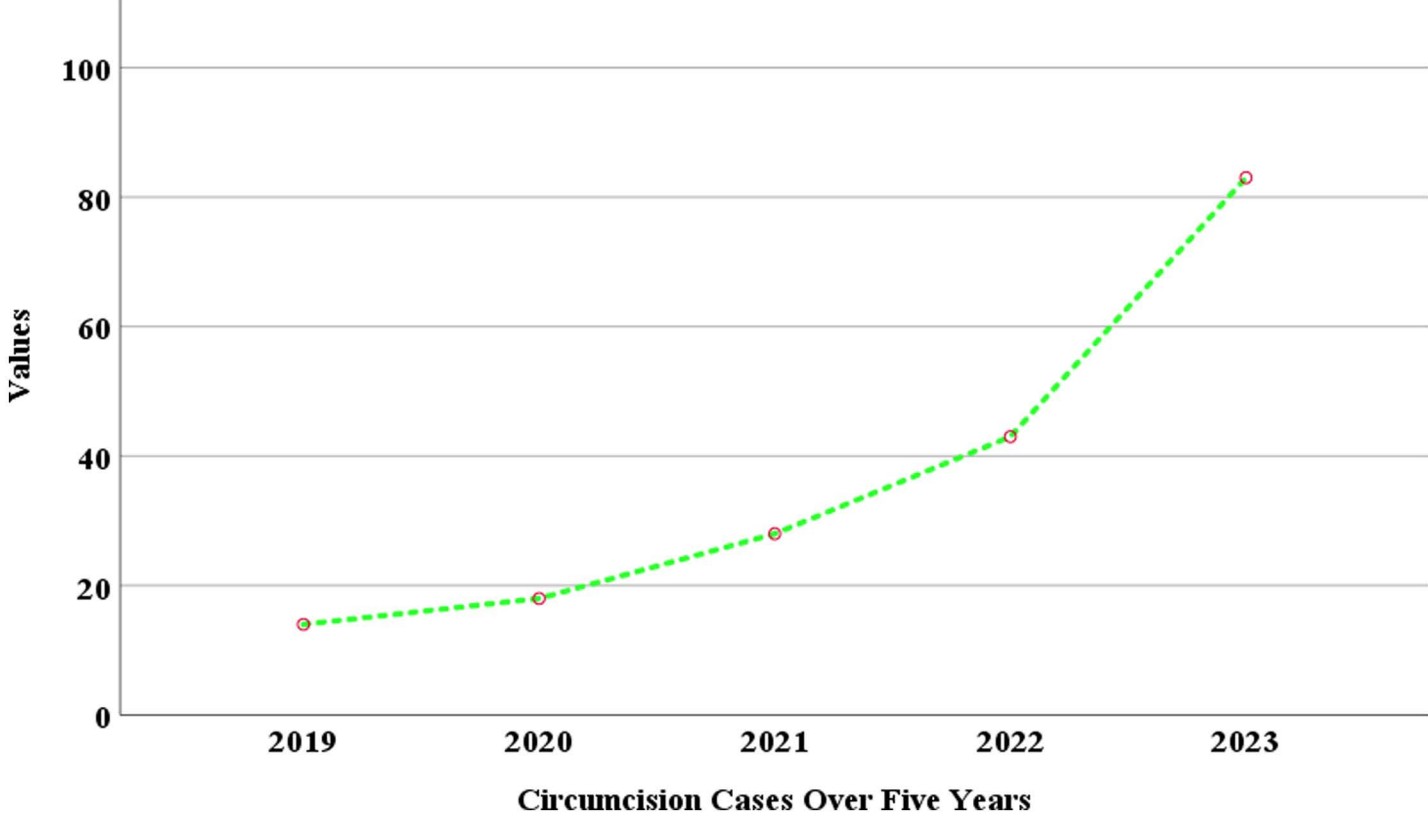

**Fig 2. Trends of circumcisions performed over the past 5 years.** A retrospective analysis of circumcision cases over a five-year period at the teaching hospital. Source: Field Data, 2024 (S1 Data).

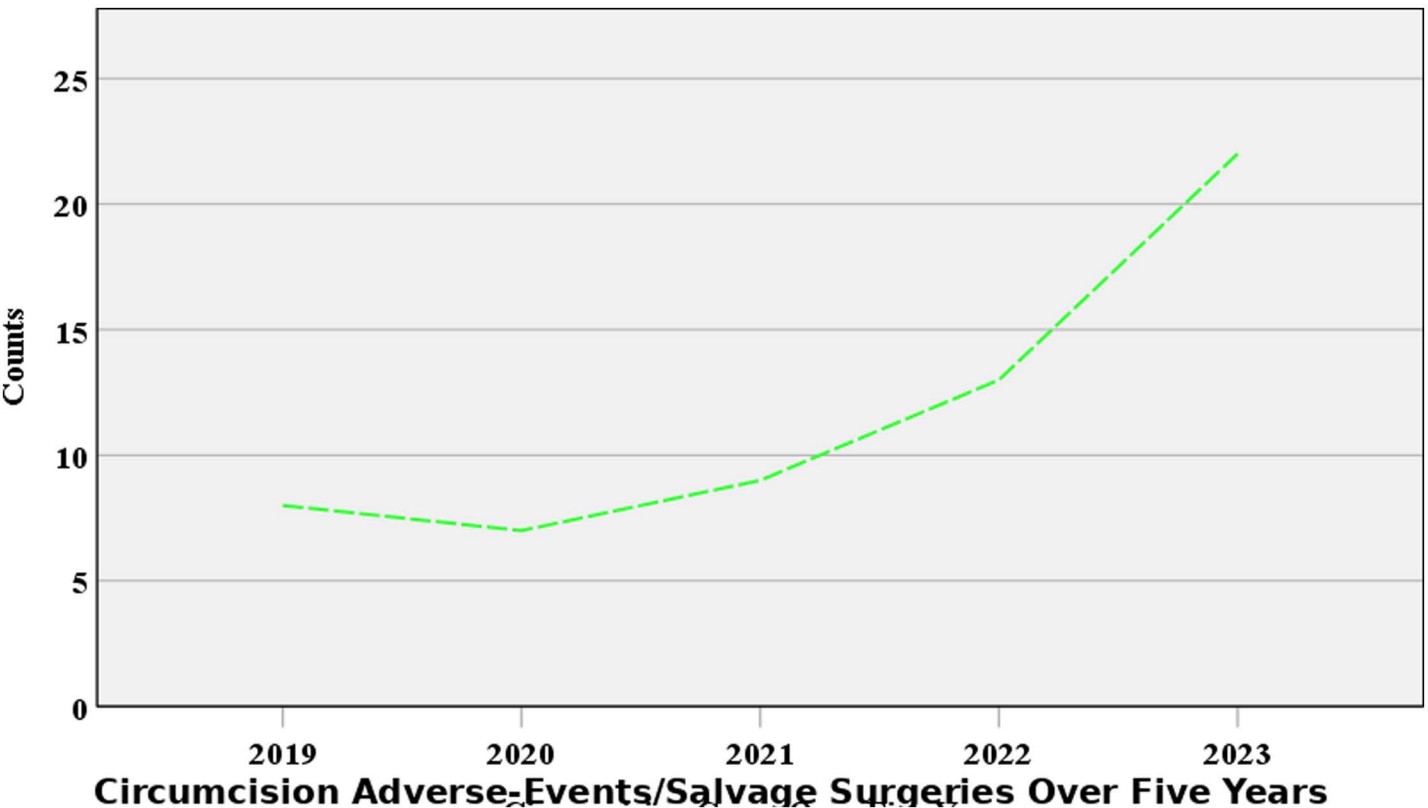

**Fig 3. Trends of salvage surgeries performed for referred circumcision mishaps over the past 5 years.** Analysis of salvage surgeries conducted for circumcision mishaps referred to the teaching hospital over the past five years. Source: Field Data, 2024 (S1 Data).

lower numbers of cases of circumcisions in the Teaching Hospital compared to the same for the entire study area suggests that most inhabitants do not see the Teaching hospital as their preferred point of call for circumcisions. It could be a pointer that participants of the study opted for non-skilled circumcisions over skilled circumcisions; and that could be a major public health bother.

### Identification of common circumcision complications: clinical parameters

**Haemoglobin levels.** Among the 44 patients (see Table 1) with circumcision complications studied, 24 (12.9%) were found to have low haemoglobin levels, which is below the normal threshold of 13.8 g/dL. The mean haemoglobin level was reported as 12.53 g/dL with a standard deviation of 2.94. Normal haemoglobin levels were observed in 16 patients (8.6%), while no patients had elevated levels). Additionally, 4 data items (2.2%) were not recorded, potentially affecting the completeness of the data.

**White blood cell counts.** The distribution of White Blood Cell (WBC) counts among the study participants revealed that no patients had low WBC counts (<4,000 μL). The mean WBC count was 1418 μL with a standard deviation of 266.88. However, 35 patients (18.8%) had normal WBC counts within the range of 4,000–11,000 μL, while 5 patients (2.7%) exhibited high WBC counts (>11,000 μL). Again, 4 records (2.2%) were not found.

**Platelet counts.** For platelet counts, 4 patients (2.2%) had low platelet levels (<150,000 μL), with a mean of **296.59, and a standard deviation of 143.36.**

**Table 1. Clinical parameters of patients with circumcision complications.**

| Variables | | Frequency (N=44) | Percent (%) | Mean | S D |
|---|---|---|---|---|---|
| **Hemoglobin Level** | Low (<13.8 g/dL) | 24 | 12.9% | **12.53** | **2.94** |
| | Normal (13.8–17.2 g/dL) | 16 | 8.6% | | |
| | High (>17.2 g/dL) | 0 | 0.0% | | |
| | Not recorded | 4 | 2.2% | | |
| **White Blood Cell** | Low (<4,000 μL) | 0 | 0.0% | **1418** | **26.88** |
| | Normal (4,000–11,000 μL) | 35 | 18.8% | | |
| | High (11,000 μL) | 5 | 2.7% | | |
| | Not recorded | 4 | 2.2% | | |
| **Platelet Count** | Low (<150,000 μL) | 4 | 2.2% | **296.59** | **143.36** |
| | Normal (150,000–450,000 μL) | 33 | 17.7% | | |
| | High (>450,000 μL) | 3 | 1.6% | | |
| | Not recorded | 4 | 2.2% | | |

*Source:* Field Data, 2024 (S1 Data).

In contrast, 33 patients (17.7%) had normal platelet counts within the range of 150,000–450,000 μL, and 3 patients (1.6%) had high platelet counts (>450,000 μL). The 4 cases (2.2%) where platelet counts were not recorded.

## Correlation between presentation types and duration of symptoms

Table 2 reveals that all 10 cases classified as acute emergencies (Trauma Early Warning Sign (TEWS)score 4-9) presented within the day of symptom onset, indicating no delays in seeking immediate care. In contrast, 1 acute emergency reported after a day. For cold presentations, all 175 cases (TEWS 0-3) were seen after a day. Additionally, the Chi-Square test result of 150.74 with a p-value of **1.2e$^{-34}$** indicates an extremely strong statistically significant association between the type of presentation (Acute Emergency or Cold Presentation) and the duration of symptoms. This may suggest once there is a complication mishap at play, the health seeking behaviour amongst the studied population is good.

## Circumcision practices and referral dynamics

Table 3 provides insights into the circumcision details including:

**Location of Circumcision**: Most circumcisions were conducted in Ho and its environs, accounting for 130 cases, which is 69.9% of the total. Nearby towns and districts recorded 16 cases (8.6%), while other regions accounted for 40 cases.

**Table 2. Analysis of presentation type and duration of symptoms in circumcision complications.**

| Variables | | Duration | | | | |
|---|---|---|---|---|---|---|
| | | Within the Day | After a Day | Total | Chi-Square | Test |
| | | N | N | N | X² | P-Value |
| **Presentation** | **Acute Emergency (TEWS 4-9)** | 10 | 1 | **11** | 150.74 | 1.2e$^{-34}$ |
| | **Cold Presentation (TEWS 0-3)** | 0 | 175 | **175** | | |
| | *Total* | **10** | **176** | **186** | | |

*Source:* Field Data, 2024 (S1 Data).

**Table 3. An analysis of location of circumcision, performers of circumcision, and decision-makers for referral when a circumcision complication occurs.**

| Variables | | Frequency | Percent (%) |
|---|---|---|---|
| **Location of Circumcision** | Ho and Surroundings | 130 | 69.9% |
| | Nearby Towns and Districts | 16 | 8.6% |
| | Other Regions | 40 | 21.5% |
| | **Total** | **186** | **100%** |
| **Performer of Circumcision** | Doctors*** | 30 | 16.1% |
| | Midwife | 72 | 39.7% |
| | Nurse | 74 | 39.8% |
| | Wanzam | 10 | 5.4% |
| | **Total** | **186** | **100%** |
| **Referral Decision-Maker** | Circumciser | 7 | 15.9% |
| | Parents | 28 | 63.6% |
| | Other Relatives | 9 | 20.5% |
| | **Total** | **44** | **100%** |

***Represents specialists, consultants, and medical doctors.

*Source:* Field Data, 2024 (S1 Data)

**Performer of Circumcision**: Out of the total cases, doctors, including specialists, consultants, and medical doctors, were responsible for 30 circumcisions, accounting for 16.1% of the total. Midwives and nurses were the most frequent providers, each performing 72 circumcisions, which represents 38.7% of the total cases. Wanzams representing 10, accounting for 5.4%.

Referral Decision-Maker: The decision-making process for referrals once a circumcision mishap occurs, is varied. 'Other Relatives' were the decision-makers in 9 cases, representing 20.5% of the total. Circumcisers made the referral decisions in 7 (15.9%) cases, and most referral decisions were taken by parents, i.e., 28 referrals, representing 63.6%.

## Encountered circumcision complications

Fig 4 categorises the findings. Out of 186 cases, 163 (87.63%) were classified as having normal examination results. Conversely, 23 cases (**12.37%**) were identified as having pathological findings.

Table 4 reveals a diverse range of conditions, each with varying prevalence. 163 cases presented with normal findings (91.9%). Hypospadias (wrongfully circumcised) was the second most frequent diagnosis, recorded in 10 cases (5.4%). Phimosis, characterized by an inability to retract the foreskin, was observed in 2 cases (1.1%). Other conditions included Urethral Meatal Stenosis (4.5%), Scrotal laceration and Infected Smegma, each reported in 1 case (0.5%). Chordae, a condition where fibrous bands form on the penis leading to clinically observable abnormal penile curvature or twist, was also noted in 1 case (0.5%).

From the findings of this study, the percentage of circumcisions done by each provider, and the percentage of complications attributable to each provider are summarised by Table 5.

**Explanation of table columns (Tables 6 and 7):**

- **Total Circumcisions**: The total number of circumcisions performed by each provider.

- **Complications (Yes)**: The number of circumcisions that resulted in complications.

- **'No Complications'**: The number of circumcisions that did not result in complications.

- **Odds of Complication**: The odds of complications occurring for each provider (Complications ÷ No Complications).

- **Odds Ratio vs Doctors**: The odds ratio comparing each provider to doctors (baseline).

- **ln(OR)**: The natural logarithm of the odds ratio.

- **SE(ln(OR))**: The standard error of the natural logarithm of the odds ratio.

- **Z-statistic**: The Z-score calculated using the odds ratio and standard error.

- **p-value**: The p-value corresponding to the Z-statistic, indicating the statistical significance ($p < 0.05$), or otherwise.

- This analysis shows that the odds of complications for circumcisions performed by **Wanzams** are statistically significantly higher than for doctors ($p < 0.05$), whereas the odds for **midwives** and **nurses** are not statistically significant compared to doctors.

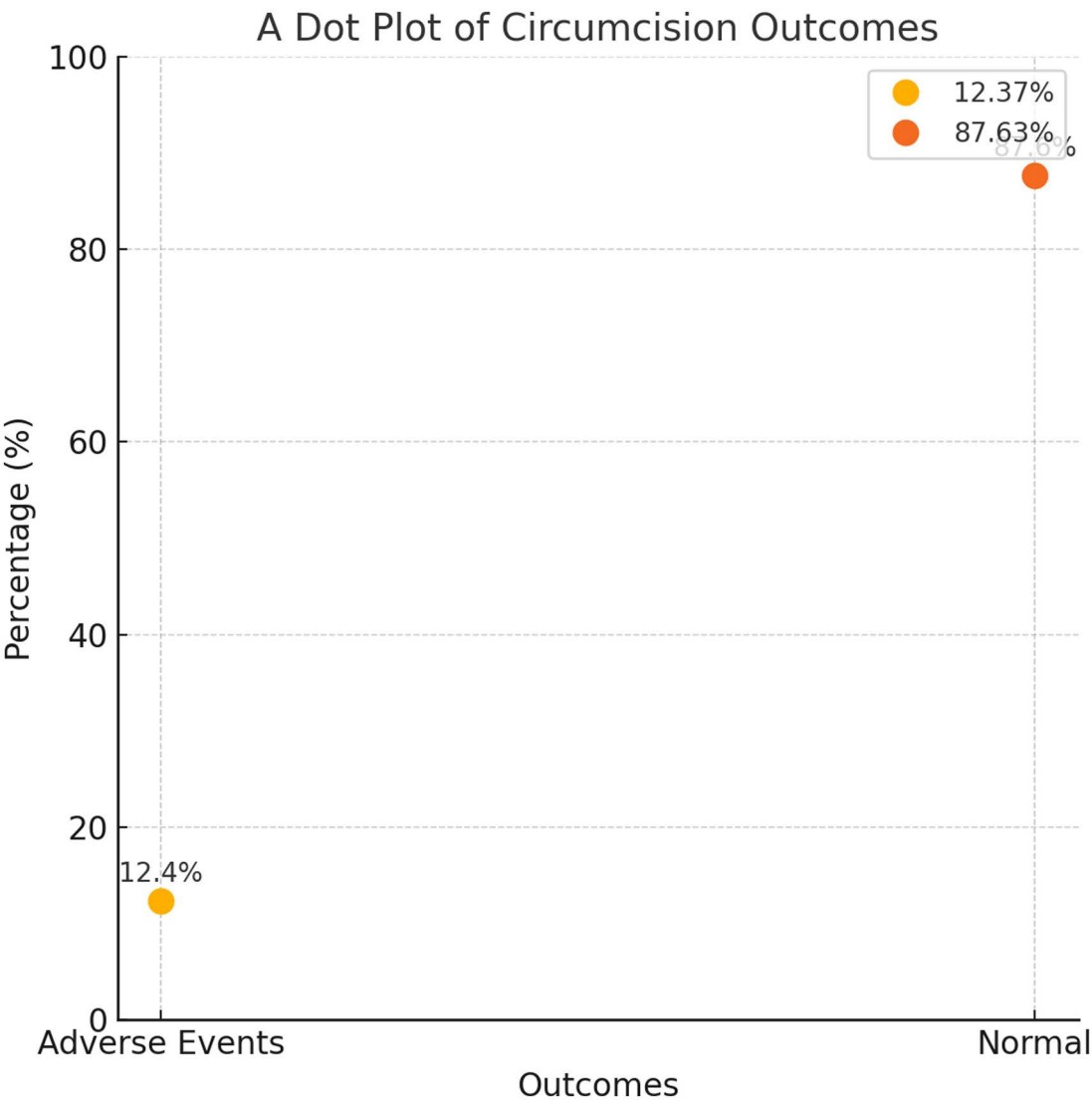

**Fig 4. Examination findings.** Findings from examination of circumcision cases, distinguishing between pathological and normal cases. Source: Field Data, 2024 (S1 Data).

**Table 4.  Reasons for circumcision and associated diagnoses in clients with adverse events.**

| Variables | Associated Diagnosis | | | |
|---|---|---|---|---|
| | Frequency | Percent | Valid Percent | Cumulative Percent |
| Chordae | 1 | .5 | .5 | .5 |
| Hypospadias (that were wrongfully circumcised outside the hospital) | 10 | 5.4 | 5.4 | 5.9 |
| Infected Smegma | 1 | .5 | .5 | 6.5 |
| Scrotal Laceration | 1 | .5 | .5 | .5 |
| None | 163 | 87.4 | 87.4 | 94.4 |
| Phimosis | 2 | 1.1 | 1.1 | 95.5 |
| Urethral Meatal Stenosis/redundant prepuce/ bridging skin | 8 | 4.5 | 4.5 | **100.0** |
| **Total** | **186** | **100.0** | **100.0** | |

*Source:* Field Data, 2024 (S1 Data).

**Table 5.  Frequency of circumcision and frequency of complications per each provider.**

| Healthcare Provider | Circumcisions Performed (%) | Complications (%) |
|---|---|---|
| Midwives | 38.2 | 13.04 |
| Nurses | 40.3 | 39.1 |
| Doctors | 16.13 | 3.4 |
| Wanzams | 5.38 | 34.8 |

Given that the total number of circumcisions in this study was 186; and the total number of circumcision complications was 23; the actual numbers are shown in the table below, as a confusion matrix table. *Source:* Field Data, 2024 (S1 Data).

## Management of complications

Fig 5 reveals that among the complicated cases of circumcisions analysed, at least, one procedure was performed as salvage surgery, with a total of 31 salvage procedures being done for the 23 circumcision mishaps. An average of 1.3 procedures per recorded circumcision complication. Revision of circumcision, Urethroplasty/ hypospadias repair, for patients with urethrocutaneous fistulae or wrongfully circumcised hypospadias, was done in 13% (7 cases). Suturing of wound was done in 5.5% (3 cases). Debridement was done in 5.5% (3 cases), while fistulectomy plus primary repair was needed in 7% (4 cases). Glansplasty and meatotomy/meatoplasty, were done in 2% (1 case) and 4% (2 cases) respectively. The patient with a chordae needed a penile degloving plus release of chordae (without Nesbit's procedure), in addition to a meatoplasty.

## Discussion

### Prevalence of circumcision complications

The study revealed a complication rate of 8.1% among circumcision 186 cases studied. This aligns closely with findings from rural Ghana (1) [10], where the complication rate is also around 8.1%. This rate is significantly higher than the <0.5% complication rate reported in the U.S. [9]. This discrepancy underscores the impact of regional factors on complication rates, as noted in the literature [5]. The higher prevalence in less developed regions such as rural Ghana and the Volta Region may be attributed to factors including low availability of

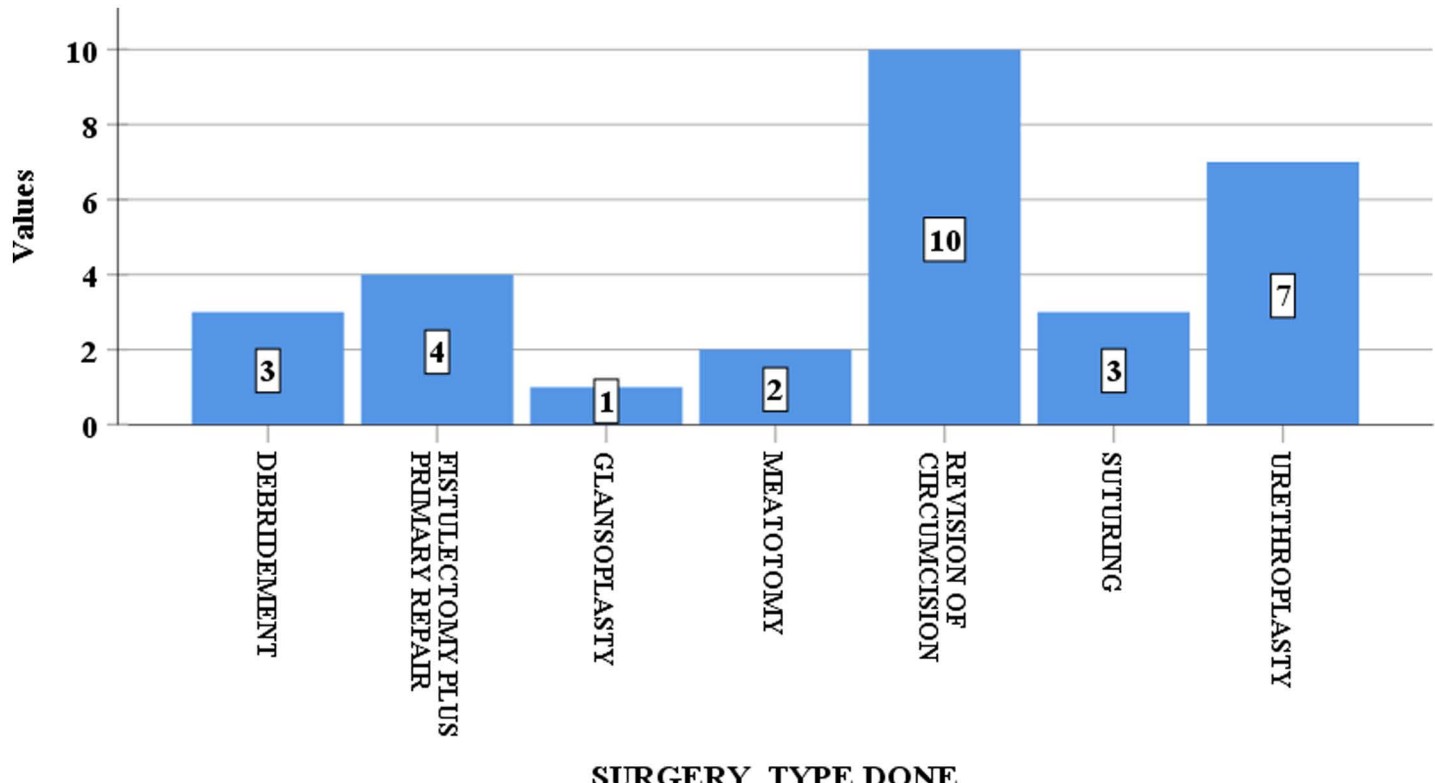

**Fig 5. Management of Complications.** Types of salvage surgeries performed to address circumcision adverse events. Source: Field Data, 2024 (S1 Data).

advanced surgical techniques, poor health seeking behaviour by the parents of these children, variable provider experience, and differing healthcare infrastructure [8].

It is disheartening to know that cases in which neonates born with hypospadias were circumcised without recourse to referral to the urologist or paediatric surgeon were rife; and seven [7] of such were encountered. The surgical principle is that until such a time that the hypospadias is going to be repaired for the neonate born with it, the prepuce should be preserved since it may serve as a source of a flap for the procedure of hypospadias repair itself [16]. This hypospadias repair procedure that utilizes a preputial flap is called the **Duckett repair** or **Duckett onlay island flap urethroplasty** [16]. This technique as posited by the sentinel article by Duckett JW(1980, 1981), involves using a flap of tissue from the inner layer of the prepuce (foreskin) to reconstruct the urethra in cases of hypospadias, particularly when the defect is more proximal. The preputial tissue provides a well-vascularized flap that helps in forming a new urethra, reducing the risk of fistula formation and other complications [16].

## Common circumcision complications

The most common complications observed in this study include infection, meatal stenosis, and bleeding. These findings are consistent with global patterns where bleeding is often the most prevalent complication [11]. In Ghana, the prevalence of urethrocutaneous fistula, as reported by Appiah et al. (2016), contrasts with our findings, where this complication was less common [17]. This variation may be attributed to differences in circumcision practices and post-operative care between regions. Our results suggest a higher incidence of infection and meatal stenosis, which could be related to local practices and the use of traditional versus

modern techniques. This aspect of our study's results is in line with Appiah et al. (2016), who reported similar complications in Ghanaian settings but with different frequencies [17].

## Relationship between provider and outcome

This study found that circumcision complications were associated with procedures performed by nurses and wanzams. Among the 186 cases analyzed, 38.2% of circumcisions were carried out by midwives, 40.3% by nurses, 16.13% by doctors and lastly 5.38% by wanzams. The complications seen was 13.04% by midwives and 3.4% by doctors, 39.1% by Nurse and 34.8% by wanzams. A significant relationship was found between the circumcision provider and complication rates (Chi-square = 16.975, p = 0.00).

The data (Tables 6 and 7) show that circumcisions in the Volta Region are performed by midwives and nurses, with a smaller proportion by traditional circumcisers and doctors. This distribution is similar to findings in other parts of Ghana, where midwives and nurses perform the majority of procedures [12,13]. The high complication rates associated with traditional circumcisers, as highlighted by Kacker & Tobian (2013), reflect the challenges of inadequate training and resources [12,13]. However, our study also identifies notable complications in procedures performed by healthcare professionals, indicating that provider experience and training remain critical factors regardless of the practitioner's background. This finding aligns with the literature suggesting that both traditional and medical circumcision practitioners in Ghana have been associated with complication rates [2].

Further analysis of complications to find the true relationship with specific circumcizors across board, revealed significant variations in risk compared to doctors. Midwives had an odds ratio (OR) of 1.278, but with a Z-statistic of 0.209 and a p-value of 0.834, indicating no statistically significant difference from the risk for doctors. Nurses exhibited a higher odds ratio of 3.954, but this higher odds of risk for circumcision mishaps also did not reach statistical significance (Z-statistic of 1.276; p-value of 0.202). In contrast, Wanzams had a

Table 6. Confusion matrix table for circumcision complications.

| Provider | Complications (Yes) | No Complications | Total Circumcisions |
|---|---|---|---|
| Midwives | 5 | 66 | 71 |
| Nurses | 9 | 66 | 75 |
| Doctors | 1 | 29 | 30 |
| Wanzams | 8 | 2 | 10 |
| Total | 23 | 163 | 186 |

Source: Field Data, 2024 (S1 Data)

Table 7. A consolidated table that captures all the key steps, calculations, and results for the odds ratios, z-statistics, and p-values for each circumcision provider compared to doctors.

| Provider | Total Circumcisions | Complications (Yes) | No Complications | Odds of Complication | Odds Ratio vs Doctors | ln(OR) | SE(ln(OR)) | Z-statistic | p-value |
|---|---|---|---|---|---|---|---|---|---|
| Doctors | 30 | 1 | 29 | 0.0345 | Baseline | Baseline | Baseline | Baseline | Baseline |
| Midwives | 71 | 3 | 68 | 0.0441 | 1.278 | 0.2457 | 1.1757 | 0.2090 | 0.834 |
| Nurses | 75 | 9 | 66 | 0.1364 | 3.954 | 1.375 | 1.0774 | 1.276 | 0.202 |
| Wanzams | 10 | 8 | 2 | 4.0000 | 115.942 | 4.753 | 1.2882 | 3.691 | 0.0002 |

Source: Field Data, 2024 (S1 Data)

dramatically elevated odds ratio of 115.942, with a Z-statistic of 3.691 and a highly significant p-value of 0.0002, reflecting an increased likelihood of complications compared to doctors (see Tables 5 to 7). These findings suggest that traditional providers, such as Wanzams, may pose a much higher complication risk during circumcision, warranting targeted interventions to improve outcomes.

## Conclusion

### The study provides valuable insights into circumcision practices and outcomes in the Volta Region

Under less-trained hands, circumcision could be catastrophic. Salvage surgeries for circumcision mishaps are associated with less favourable outcomes in up to one-third of the cases, suggesting that circumcision mishaps are better prevented than cured/salvaged.

The increasing number of circumcision cases, coupled with the identification of common complications and variations in practice, highlights the need for improved standards, training, and public education.

To improve circumcision practices in the Volta Region, the following recommendations are proposed. First, healthcare provider training should be standardized, ensuring practitioners are skilled in surgical techniques and post-operative care. Regulations must restrict circumcision to qualified individuals, requiring certification and continuing education. Public awareness campaigns should educate parents and caregivers on the benefits and risks of circumcision, emphasizing the need for circumcisions by qualified practitioners, using local languages and community outreach. Also, Public health campaigns to dissuade non-surgeon circumcizors to refrain from circumcising children with hypospadias but refer them are urgently needed.

A comprehensive system should be developed by the health system to monitor circumcision cases and outcomes, helping identify trends and improving care. Facilities should maintain detailed records, including pre- and post-operative care. Infection control protocols must be enforced by the health system, with practitioners trained in sterilization, aseptic techniques, and early complication management.

Collaboration between traditional and modern healthcare providers should be encouraged, ensuring safe practices while respecting cultural traditions. Training for traditional practitioners on safety and complication management is essential. Lastly, strong referral systems should be established by the health system to ensure timely care for complications, with clear protocols for specialist referrals.

## Supporting information

**S1 Data. Demographic and clinical characteristics of circumcision cases over a five-year period (field data, 2024). Description:** This dataset contains detailed demographic, clinical, and circumcision-related characteristics of patients from a teaching hospital in Ghana. It includes variables such as age, platelet count, presentation type (acute or cold), complaints, duration of complaints, and the facility or circumciser associated with the procedure. The data covers cases spanning a five-year retrospective analysis. This file is included as a supporting information file to provide context and detail for the findings discussed in the manuscript. The data underscores the patterns of complications and clinical presentations of circumcision cases in the study setting. (XLSX)

## Acknowledgment

We acknowledge the authorities of the Teaching Hospital.

## Author contributions

**Conceptualization:** frank obeng, Sylvester Appiah Boakye.

**Data curation:** Frank Obeng, Sylvester Appiah Boakye.

**Formal analysis:** Frank Obeng, Sylvester Appiah Boakye.

**Investigation:** Frank Obeng, Sylvester Appiah Boakye.

**Methodology:** Frank Obeng, Sylvester Appiah Boakye.

**Project administration:** Frank Obeng, Sylvester Appiah Boakye.

**Resources:** Frank Obeng, Sylvester Appiah Boakye.

**Software:** Frank Obeng, Sylvester Appiah Boakye.

**Supervision:** Frank Obeng.

**Validation:** Frank Obeng.

**Visualization:** Frank Obeng, Sylvester Appiah Boakye.

**Writing – original draft:** Frank Obeng, Sylvester Appiah Boakye.

**Writing – review & editing:** Frank Obeng.

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
