## [Decision Letter · Decision Letter 0]

5 Nov 2024

PGPH-D-24-02131

Clinical Outcomes and Prevalence of Complications of Male Circumcisions: A Five-Year Retrospective Analysis at a Teaching Hospital in Ghana

Dear Dr. OBENG,

Thank you for submitting your manuscript to PLOS Global Public Health. After careful consideration, we feel that it has merit but does not fully meet PLOS Global Public Health’s publication criteria as it currently stands. Therefore, we invite you to submit a revised version of the manuscript that addresses the points raised during the review process.

Please note that we have only been able to secure a single reviewer to assess your manuscript. We are issuing a decision on your manuscript at this point to prevent further delays in the evaluation of your manuscript. Please be aware that the editor who handles your revised manuscript might find it necessary to invite additional reviewers to assess this work once the revised manuscript is submitted. However, we will aim to proceed on the basis of this single review if possible. 

We look forward to receiving your revised manuscript.

Kind regards,

Steve Zimmerman, PhD

PLOS Staff Editor

Journal Requirements:

 1. After internal review, we do not feel that the images included in your submission are necessary to answer your research question or support your results. We therefore suggest that they be removed. 2. In the online submission form, you indicated that "DATA WILL BE AVAILABLE UPON REQUEST".  All PLOS journals now require all data underlying the findings described in their manuscript to be freely available to other researchers, either 1. In a public repository, 2. Within the manuscript itself, or 3. Uploaded as supplementary information. This policy applies to all data except where public deposition would breach compliance with the protocol approved by your research ethics board. If your data cannot be made publicly available for ethical or legal reasons (e.g., public availability would compromise patient privacy), please explain your reasons by return email and your exemption request will be escalated to the editor for approval. Your exemption request will be handled independently and will not hold up the peer review process, but will need to be resolved should your manuscript be accepted for publication. One of the Editorial team will then be in touch if there are any issues. 3. Please provide separate figure files in .tif or .eps format. For more information about figure files please see our guidelines:  https://journals.plos.org/globalpublichealth/s/figures https://journals.plos.org/globalpublichealth/s/figures#loc-file-requirements  4. Please provide an Author Summary. This should appear in your manuscript between the Abstract (if applicable) and the Introduction, and should be 150–200 words long. The aim should be to make your findings accessible to a wide audience that includes both scientists and non-scientists. Sample summaries can be found on our website under Submission Guidelines:  https://journals.plos.org/globalpublichealth/s/submission-guidelines#loc-parts-of-a-submission

Additional Editor Comments (if provided):

Reviewers' comments:

Reviewer's Responses to Questions

**Comments to the Author**

1. Does this manuscript meet PLOS Global Public Health’s publication criteria ? Is the manuscript technically sound, and do the data support the conclusions? The manuscript must describe methodologically and ethically rigorous research with conclusions that are appropriately drawn based on the data presented.

Reviewer #1: Yes

2. Has the statistical analysis been performed appropriately and rigorously?

Reviewer #1: Yes

3. Have the authors made all data underlying the findings in their manuscript fully available (please refer to the Data Availability Statement at the start of the manuscript PDF file)?

Reviewer #1: Yes

4. Is the manuscript presented in an intelligible fashion and written in standard English?

Reviewer #1: Yes

5. Review Comments to the Author

Reviewer #1: Summary

The authors reviewed five-year MC data from Ho Teaching Hospital in in the Volta region of Ghana to determine the clinical outcomes of circumcisions and associated adverse events (AE). The AE rate was 12.37% (23/186) with incomplete removal of the foreskin as the most common adverse outcome (43.48%) followed by post-circumcision bleeding at 21.74%. Circumcisions conducted by doctors and traditional practitioners were associated with fewer AE’s than those conducted by nurses and overall success rate for corrective surgeries following circumcision AES was 70%. They emphasized the need for training, guidance, and policy interventions to reduce the incidence of circumcision-related AEs.

General Comments

The manuscript is clearly written and addresses the important topic of adverse events associated with male circumcision. It could however benefit from some edits.

Consider replacing the terms mishap, complications, and catastrophe with adverse events in describing the unfavorable outcomes of circumcision throughout the manuscript.

The focus of this paper appears to be on circumcisions conducted in the neonatal period or early infancy. Consider editing the title to reflect this focus. The results section requires focused attention.

Abstract:

Please include geographic context to the opening statement:…with most circumcisions occurring during the neonatal period. Specify whether this statement applies globally or regionally.

Introduction:

Prevalence of circumcision- If possible, consider presenting background information on the prevalence circumcision related AEs by the varying contents in which the procedure is conducted e.g. therapeutic circumcisions for conditions of the foreskin, Voluntary Medical Male circumcision for HIV prevention and traditional circumcision.

Materials and Methods

The study population included circumcised children attended at HOTH and referrals with AEs

from other health facilities within the Volta region. The AE prevalence estimate based on this study may be inflated because referrals from other facilities was limited to those who experienced AEs. Those without AEs we excluded from the denominator. This may be stated as a limitation.

Ethical statement

Given that this was a retrospective based on review of data collected over the previous five years, pleas details of how consent was obtained for clients whose clinical pictographs are included in the manuscript.

Results:

Overall, the results section requires considerable edits to align with the key objectives of the manuscript and to improve clarity.

Fig 1: Trends of Circumcisions Performed Over the Past 5 Years

Please confirm if the reported numbers are for Ho Teaching Hospital alone or whether they include the referring facilities within its catchment area.

Figure 2: Trends of Circumcisions and Salvage Surgeries for Referred Circumcision Mishaps Performed Over the Past 5 Years within the Teaching Hospital

The graph only appears to present the trend in corrective surgeries conducted at Teaching Hospital (59), yet the title suggests two variables (including Trend of Circumcisions). Kindly review.

Additionally, review the associated narrative for this graph to improve clarity for the readers. The key message is not clear to me.

Table 1: Clinical Parameters of Patients with Circumcision Complications

It is stated in the abstract that among 186 circumcision cases, 23 (12.37%) experienced complications. Table 1 one suggests that there were 44 patients with circumcision complications. I may be missing some details. Given that 44 corrective surgeries were conducted among 23 clients with adverse events, is it possible that the measurements in table 1 were taken around the time of each procedure? This would result in multiple measurements for some individuals.

Table 2: Analysis of Presentation Type and Duration of Symptoms in Circumcision

Complications

Include in a summary of the framework used to classify circumcision-related AEs as acute or cold (either within the manuscript or as an appendix). Consider revising the table caption to reflect the key message in the table. Comparison time lapse before of seeking for clients experiencing acute vs cold symptoms of circumcision-related AEs. Key message appears to be: 9 out 10 clients with acute symptoms presented for care within the same day of symptoms onset. All the 175 clients classified as cold cases presented for care one day after the onset of symptom. Please review to improve clarity for readers.

Table 3 An Analysis of Location of circumcision, Performers of circumcision, and

Decision-Makers for Referral when a circumcision complication occurs.

Based on Referral Decision-Makers, there were 44 referrals, yet there were only 23 unique individuals with complications. Does it mean that some individuals who had multiple AEs were referred by different individuals for each AE?

Table 4: Associated Diagnoses in Circumcision Complications

This table title should be edited to improve its clarity. The title appears to suggest a presentation of additional coincidental diagnoses among clients who had circumcision related adverse events. If that is the case, then the denominator should be 23. It is not clear why 163 uneventful procedures with no complications are included in this table.

Otherwise, it appears that reasons for circumcisions were combined with adverse events associated with the procedure plus coincidental conditions found in circumcised clients. This may confuse readers.

Merge Table 5 (Frequency of Circumcision and Frequency of Complications per each

Provider) and Table 6 (Confusion Matrix Table for Circumcision Complications) and edit caption.

6. PLOS authors have the option to publish the peer review history of their article (what does this mean? ). If published, this will include your full peer review and any attached files.

**Do you want your identity to be public for this peer review?** For information about this choice, including consent withdrawal, please see our Privacy Policy .

Reviewer #1: No

---

## [Decision Letter · Decision Letter 1]

9 Jan 2025

Clinical Outcomes and Prevalence of Complications of Male Circumcisions: A Five-Year Retrospective Analysis at a Teaching Hospital in Ghana

PGPH-D-24-02131R1

Dear DR OBENG,

We are pleased to inform you that your manuscript 'Clinical Outcomes and Prevalence of Complications of Male Circumcisions: A Five-Year Retrospective Analysis at a Teaching Hospital in Ghana' has been provisionally accepted for publication in PLOS Global Public Health.

Best regards,

Atanu Bhattacharjee, Ph.D

Academic Editor

Reviewer Comments (if any, and for reference):

Reviewer's Responses to Questions

**Comments to the Author**

1. If the authors have adequately addressed your comments raised in a previous round of review and you feel that this manuscript is now acceptable for publication, you may indicate that here to bypass the “Comments to the Author” section, enter your conflict of interest statement in the “Confidential to Editor” section, and submit your "Accept" recommendation.

Reviewer #1: All comments have been addressed

2. Does this manuscript meet PLOS Global Public Health’s publication criteria ? Is the manuscript technically sound, and do the data support the conclusions? The manuscript must describe methodologically and ethically rigorous research with conclusions that are appropriately drawn based on the data presented.

Reviewer #1: Yes

3. Has the statistical analysis been performed appropriately and rigorously?

Reviewer #1: Yes

4. Have the authors made all data underlying the findings in their manuscript fully available (please refer to the Data Availability Statement at the start of the manuscript PDF file)?

Reviewer #1: Yes

5. Is the manuscript presented in an intelligible fashion and written in standard English?

Reviewer #1: Yes

6. Review Comments to the Author

Reviewer #1: No new comments.

7. PLOS authors have the option to publish the peer review history of their article (what does this mean? ). If published, this will include your full peer review and any attached files.

**Do you want your identity to be public for this peer review?** For information about this choice, including consent withdrawal, please see our Privacy Policy .

Reviewer #1: No
